# Perusing Buchbinder–Lyakhovich Canonical Formalism for Higher-Order Theories of Gravity

**Dalia Saha** [1,2] and **Abhik Kumar Sanyal** [2,*]

1   Department of Physics, University of Kalyani, Kalyani 741235, West Bengal, India
2   Department of Physics, Jangipur College, Murshidabad 742213, West Bengal, India
*   Correspondence: sanyal_ak@yahoo.com

**Abstract:** Ostrogradsky's, Dirac's, and Horowitz's techniques in terms of higher-order theories of gravity produce identical phase-space structures. The problem with these techniques is manifested in the case of Gauss–Bonnet–dilatonic coupled action in the presence of higher-order term, in which case, classical correspondence cannot be established. Here, we explore another technique developed by Buchbinder and their collaborators (BL) and show that it also suffers from the same disease. However, when expressing the action in terms of the three-space curvature, and removing 'the total derivative terms', if Horowitz's formalism or even Dirac's constraint analysis is pursued, all pathologies disappear. Here, we show that the same is true for BL formalism, which appears to be the simplest of all the techniques to handle.

**Keywords:** higher-order theory; canonical formulation

## 1. Introduction

The canonical formulation of higher-order theories was developed by Ostrogradsky almost two centuries ago [1,2]. However, it did not draw much attention, since other than toy mechanical models, practically no such physical theories were proposed at that time. Exactly a century had elapsed when it was applied to a physically motivated problem, such as a fourth-order harmonic oscillator [3]. The real physical problem in this context appeared for the first time, while an attempt was made to formulate a renormalized quantum theory of gravity [4]. The higher-derivative theory of gravity is usually considered a model of quantum gravity. The reason is that Einstein–Hilbert action is supplemented by curvature squared terms ($R^2$, $R_{\mu\nu}R^{\mu\nu}$) to ensure renormalizability [4] and asymptotic freedom [5–7]. Unfortunately, curvature-squared gravity theories have been found to suffer from the unresolved problem of physical unitarity in perturbative analysis, which is typical for higher-derivative theories. However, possibilities to overcome this difficulty were also discussed in the literature [6,8] and references therein. It is also ascertained that curvature squared gravity would arise as a low-energy effective theory derived from superstring theory in $D = 10$ dimensions [9–11]. Over the last couple of decades, higher-order theories of gravity, e.g., $F(R)$, $F(\mathcal{G})$, $F(R,T)$, etc. ($R$, $\mathcal{G}$, $T$ being the Ricci scalar, the Gauss–Bonnet term, and the torsion term, respectively) have drawn lot of attention in the search for alternatives to the dark energy issue. Nonetheless, it is always suggested to test the viability of such modified theories of gravity in different contexts. In the context of the very early universe, a canonical formulation is required as a precursor, especially to study quantum cosmology.

Since Ostrogradsky's technique does not apply in the degenerate case of singular Lagrangian, for which the Hessian determinant vanishes, Dirac's constraint analysis [12] may be applied for the purpose. Nonetheless, a host of theories have been formulated over the years to bypass constraint analysis. One such approach in this direction was originally proposed by Boulware [13], and later reshuffled by Horowitz [14] in the context of a higher-order theory of gravity, in particular. Since the canonical formulation of higher-order

theories require an extra degree of freedom, in Horowitz's formalism, apart from the scale factor ('*a*' in the Robertson–Walker minisuperspace), an auxiliary variable is introduced by taking the derivative of the action (*A*, say) with respect to the highest derivative of the field variable present ($Q = \frac{\partial A}{\partial \ddot{a}}$). At the end, the auxiliary variable is replaced by the basic variable (extrinsic curvature tensor) through canonical transformation. The important finding in this regard is that all three formalisms, viz. Ostrogradsky's (once degeneracy is removed), Dirac's, and Horowitz's formalisms, produce an identical phase-space structure [15]. In the meantime, certain pathologies with Horowitz's formalism have been perceived. For example, it was noticed that Horowitz's formalism can even be applied in the case of linear gravity theory (Einstein–Hilbert action) leading to incorrect quantum dynamics [16–18], and some superfluous total derivative terms are eliminated [18,19], which neither may be obtained from the variational principle, nor have any connection with the Gibbons–Hawking–York term [20,21], or any of its modified versions, associated with higher-order gravity. Furthermore, the coupling parameter, in the case of the 'non-minimally coupled scalar tensor theory of gravity associated with higher-order terms', has not been found to play any particular role, since its derivative does not appear in the Hamiltonian [22]. The same is true for 'dilatonic coupled Gauss–Bonnet-theory in the presence of higher-order terms', where additional classical correspondence from the quantum counterpart cannot be established [22]. In view of such a situation, yet another technique was developed under the name 'modified Horowitz formalism' (MHF), which was successfully applied in different modified higher-order theories of gravity, to explore the evolution of the very early universe [15,17–19,22–32]. In MHF, the action is expressed in terms of the three space curvature (instead of the scale factor), 'the total derivative terms' are removed by integrating the action by parts, and Horowitz's formalism (introduction of auxiliary variable and else) is followed, thereafter.

To be specific, let us consider the following isotropic and homogeneous Robertson–Walker (RW) metric

$$ds^2 = -N^2(t)\, dt^2 + a^2(t)\left[\frac{dr^2}{1 - kr^2} + r^2(d\theta^2 + sin^2\theta d\phi^2)\right],\tag{1}$$

for which the degeneracy in the Lagrangian disappears if the gauge (*N*) is fixed a priori, and in which case, Ostrogradsky's technique applies as well. Once such degeneracy is removed, it was observed that Ostrogradsky's technique produces the same Hamiltonian obtained following Horowitz's and Dirac's formalisms [15]. Therefore, it certainly follows that both Ostrogradsky's and Dirac's formalism implicitly suffer from the same disease, in disguise, as was noticed in Horowitz's technique, discussed above. Therefore in the MHF, instead of the scale factor, the action is expressed in terms of the basic variable $h_{ij}$—the three space metric from the very beginning—so that redundant total derivative terms do not appear [18,19]. Thereafter, total derivative terms are integrated out by parts, which are canceled with the supplementary boundary (Gibbons–Hawking–York and modified Gibbons–Hawking–York) terms. Subsequently, the auxiliary variable is introduced following Horowitz's proposal. At the end, the auxiliary variable is replaced by the other basic variable $K_{ij}$—the extrinsic curvature tensor. In the process, obnoxious pathologies which appeared following Horowitz's formalism, disappear, while a different Hamiltonian is produced altogether. It is to be mentioned that although both the Hamiltonians (obtained following MHF and Ostrogradky's/Dirac's/Horowitz's formalisms) are related through canonical transformation, they indeed produce different dynamics in the quantum domain. It is also important to mention that it is not plausible to carryover the classical canonical transformations in the quantum domain for higher-order theories, due to non-linearity. MHF leads to a hermitian effective Hamiltonian, a standard quantum mechanical probabilistic interpretation, and a viable semiclassical treatment, which exhibits oscillation of the wave function with regard to the classical de-Sitter solution. As a result, classical correspondence is established. In this regard, MHF may be considered the most viable technique to handle higher-order theories. It was later established that once the action

is expressed in terms of the three-space metric ($h_{ij}$) from the very beginning, and total derivative terms are taken care of, Dirac's constraint analysis [12] also produces an identical Hamiltonian to the MHF [22,28–30].

Amongst other techniques, the Hawking–Luttrell technique [33] has limited application, since conformal transformation is not possible in general [19], and Schmidt's technique [34] is identical to Horowitz's formalism in disguise [17]. However, there is yet another technique developed in the 1980s by Buchbinder and his collaborators [35–39], which did not receive much attention. Querella [40] only noticed that although at a first glance, the general formalism developed by Buchbinder and his collaborators (BL) appears to be satisfactory, it nevertheless has pitfalls. The BL formalism is our current concern. Here, we test the abstract theoretical settings of BL formalism in a simple minisuperspace model to explore the pitfalls, if any. The underlying essence of this formalism is to bypass Dirac's constrained analysis, very much like Horowitz's technique, but instead of introducing an auxiliary variable, here, the program is initiated with the scale factor and the extrinsic curvature tensor. However, we shall, on the contrary, develop the formalism starting from the basic variables $\{h_{ij}, K_{ij}\}$, the three-space curvature, and the extrinsic curvature tensors for the reason already discussed. In our present attempt to explore the outcome of this technique, we discover that the formalism leads to an identical phase-space structure as was found in the case of Ostrogradsky's/Dirac's/Horowitz's formalism.

This paper is organized as follows. In the following section, we study the scalar tensor theory of gravity (both the minimal and non-minimal cases), and also supplement the Gauss–Bonnet–dilatonic coupled action with the scalar curvature squared ($R^2$) term, following BL formalism. In Section 3, we explore the fact that once total derivative terms are taken care of, the Hamiltonian does not differ from MHF. Section 4 discusses its physical application, in connection with some earlier works. Section 5 concludes our work.

## 2. BL Formalism in Three Different Higher-Order Theories

In view of the importance of the higher-order curvature invariant terms required to construct a renormalizable quantum theory of gravity when the curvature is extremely strong, a unique canonical formulation of the Einstein–Hilbert action supplemented by higher-order curvature invariant terms is therefore necessary. Here, we consider three different cases, minimally and non-minimally coupled scalar–tensor theory of gravity supplemented by the $R^2$ term, and the scalar–tensor theory of gravity being supplemented by $R^2$ and Gauss–Bonnet terms. In the Robertson–Walker minisuperspace (1) under consideration, the Ricci scalar and the Gauss–Bonnet terms are

$$R = \frac{6}{N^2}\left(\frac{\ddot{a}}{a} + \frac{\dot{a}^2}{a^2} + N^2\frac{k}{a^2} - \frac{\dot{N}\dot{a}}{Na}\right) \tag{2}$$

$$\mathcal{G} = R^2 - 4R_{\mu\nu}R^{\mu\nu} + R_{\alpha\beta\mu\nu}R^{\alpha\beta\mu\nu} = \frac{24}{N^3a^3}\left(N\ddot{a} - \dot{N}\dot{a}\right)\left(\frac{\dot{a}^2}{N^2} + k\right), \tag{3}$$

respectively. For the sake of comparison with earlier results, we express actions in terms of the three-space metric, instead of the scale factor, as its importance has been mentioned already, and will be explicitly shown at the beginning of Section 3. Since the construction of higher-order theory to its canonical form requires an additional degree of freedom, in addition to the three-space metric $h_{ij}$, the extrinsic curvature tensor $K_{ij}$ is treated as a basic variable, as already stated. We therefore choose the basic variables $h_{ij} = z\delta_{ij} = a^2\delta_{ij}$, so that $K_{ij} = -\frac{\dot{h}_{ij}}{2N} = -\frac{a\dot{a}}{N}\delta_{ij} = -\frac{\dot{z}}{2N}\delta_{ij}$. In terms of $z = a^2$, the Ricci scalar and the Gauss–Bonnet terms take the following forms,

$$R = \frac{6}{N^2}\left[\frac{\ddot{z}}{2z} + N^2\frac{k}{z} - \frac{1}{2}\frac{\dot{N}\dot{z}}{Nz}\right], \tag{4}$$

$$\mathcal{G} = \frac{12}{N^2}\left(\frac{\ddot{z}}{z} - \frac{\dot{z}^2}{2z^2} - \frac{\dot{N}\dot{z}}{Nz}\right)\left(\frac{\dot{z}^2}{4N^2z^2} + \frac{k}{z}\right),\tag{5}$$

respectively. It is noteworthy that since

$$R_{\mu\nu}R^{\mu\nu} = \frac{12}{N^4}\left[\frac{\ddot{a}^2}{a^2} + \frac{\dot{a}^2\ddot{a}}{a^3} + \frac{\dot{a}^4}{a^4} - 2\frac{\dot{N}\dot{a}\ddot{a}}{Na^2} - \frac{\dot{N}\dot{a}^3}{Na^3} + \frac{\dot{N}^2\dot{a}^2}{N^2a^2} + k\frac{N^2\ddot{a}}{a^3} + 2k\frac{N^2\dot{a}^2}{a^4} - k\frac{N\dot{N}\dot{a}}{a^3} + k^2\frac{N^4}{a^4}\right],\tag{6}$$

therefore,

$$R_{\mu\nu}R^{\mu\nu} - \frac{1}{3}R^2 = -\left(\frac{12}{Na^3}\right)\frac{d}{dt}\left[\frac{1}{3}\frac{\dot{a}^3}{N^3} + k\frac{\dot{a}}{N}\right],\tag{7}$$

and as a result,

$$\int\left[R_{\mu\nu}R^{\mu\nu} - \frac{1}{3}R^2\right]\sqrt{-g}d^4x = -12C\int\left[\frac{d}{dt}\left(\frac{1}{3}\frac{\dot{a}^3}{N^3} + k\frac{\dot{a}}{N}\right)\right]dt\tag{8}$$

is a total derivative term. Thus, the $R_{\mu\nu}R^{\mu\nu}$ term is redundant in the RW metric, once the $R^2$ term is considered (the constant $C$ appears due to the integration of the three spaces). Hence, to scrutinize the BL formalism presented by Buchbinder and their collaborators in the RW minisuperspace model (1), we consider scalar tensor theories of gravity and also Gauss–Bonnet–dilatonic coupled gravity theory, associated with the scalar curvature squared term $R^2$.

### 2.1. Minimal Coupling

Let us start with the following minimally coupled case,

$$A_1 = \int\sqrt{-g}\left[\alpha R + \beta R^2 - \frac{1}{2}\phi_{,\mu}\phi^{,\nu} - V(\phi)\right]d^4x + \alpha\Sigma_R + \beta\Sigma_{R^2}.\tag{9}$$

In the above, $\alpha = \frac{1}{16\pi G}$, $\beta$ is a constant coupling parameter, $\alpha\Sigma_R = 2\alpha\oint_{\partial V}K\sqrt{h}d^3x$ is the Gibbons–Hawking–York boundary term [21] associated with the Einstein–Hilbert sector of the above action, and $\beta\Sigma_{R^2} = 4\beta\oint_{\partial V}RK\sqrt{h}d^3x$ is its modified version corresponding to the $R^2$ term, while $K$ is the trace of the extrinsic curvature tensor $K_{ij}$. Note that both the counter terms are required under the condition $\delta R = 0$, at the boundary. Instead, if the condition $\delta K_{ij} = 0$ is chosen at the boundary, the counter terms are not required, as in the case of Horowitz's formalism [14], since both the boundary terms appearing under metric variation vanish. However, in the case of Ostrogradsky's technique [1] and Dirac constraint analysis [12], boundary terms are not taken care of. This is also true for BL formalism, as we shall see shortly. Nevertheless, the Modified Horowitz's formalism [17–19,23–25] fixes $\delta h_{ij} = 0 = \delta R$ at the boundary, and hence requires supplementary boundary terms. We demonstrated earlier that proper attention is paid to all boundary terms in the 'Modified Horowitz formalism' (MHF). As a result, it presents a different phase-space Hamiltonian for a particular action being supplemented by higher-order terms. Nonetheless, it is related to the others under a suitable set of canonical transformations [15]. However, as mentioned earlier, such transformations cannot be carried over in the quantum domain, due to non-linearity. So, it is indeed required to check if the BL formalism also produces the same. The action (9) in the RW minisuperspace model (1) may be written in terms of the basic variable $h_{ij} = z\delta_{ij}$ as

$$A_1 = \int\left[3\alpha\sqrt{z}\left(\frac{\ddot{z}}{N} - \frac{\dot{N}\dot{z}}{N^2} + 2kN\right) + \frac{9\beta}{\sqrt{z}}\left(\frac{\ddot{z}^2}{N^3} - \frac{2\dot{N}\dot{z}\ddot{z}}{N^4} + \frac{\dot{N}^2\dot{z}^2}{N^5} - \frac{4k\dot{N}\dot{z}}{N^2} + \frac{4k\ddot{z}}{N} + 4k^2N\right)\right.$$
$$\left. + z^{\frac{3}{2}}\left(\frac{\dot{\phi}^2}{2N} - VN\right)\right]dt + \alpha\Sigma_R + \beta\Sigma_{R^2}.\tag{10}$$

The $\binom{0}{0}$ component of the field equation in terms of the scale factor '$a$' takes the following form

$$
\begin{aligned}
\frac{6\alpha}{a^2}\left(\frac{\dot{a}^2}{N^2}+k\right)+\frac{36\beta}{a^2N^4}\Bigg(&2\dot{a}\,\dddot{a}-2\dot{a}^2\frac{\ddot{N}}{N}-\ddot{a}^2-4\dot{a}\ddot{a}\frac{\dot{N}}{N}+2\dot{a}^2\frac{\ddot{a}}{a}+5\dot{a}^2\frac{\dot{N}^2}{N^2}-2\frac{\dot{a}^3\dot{N}}{aN}\\
&-3\frac{\dot{a}^4}{a^2}-2kN^2\frac{\dot{a}^2}{a^2}+\frac{k^2N^4}{a^2}\Bigg)-\left(\frac{\dot{\phi}^2}{2N^2}+V\right)=0,
\end{aligned}
\tag{11}
$$

which contains terms up to the third derivative. This is the energy constraint Equation ($E=0$), and when expressed in terms of the phase space variables, becomes the Hamiltonian constraint equation (due to diffeomorphic invariance) of the theory under consideration. Our aim is to construct the phase-space structure and establish diffeomorphic invariance of the theory, following the formalism presented by Buchbinder and his collaborators (BL).

The Equation (9) has already been expressed in terms of the basic variable $\{h_{ij}\}$, instead of the scale factor. The canonical formulation of higher-order theories requires an additional degree of freedom, and the only choice is the the extrinsic curvature tensor $\{K_{ij}\}$. In contrast to Horowitz's formalism, where apart from $\{h_{ij}\}$ an auxiliary variable is introduced and, at the end the Hamiltonian, is expressed in terms of the basic variables $\{h_{ij}, K_{ij}\}$, in BL formalism, these basic variables are associated from the very beginning. In the Robertson–Walker metric, the extrinsic curvature tensor is expressed as,

$$
K_{ij}=-\frac{\dot{h}_{ij}}{2N}=-\frac{2a\dot{a}}{2N}\delta_{ij}=-\frac{\dot{z}}{2N}\delta_{ij}=-q_{ij}\ \text{ say.}
\tag{12}
$$

Since there is only one independent component, instead of $q_{ij}$, the new generalized coordinate is chosen to be its trace, viz.

$$
q=\frac{3\dot{z}}{2N},\ \ i.e.,\ q_{ij}=\frac{q}{3}\delta_{ij}.
\tag{13}
$$

To express the action in terms of velocities, we choose

$$
v\equiv\dot{q},\ \ v_\phi\equiv\dot{\phi}.
\tag{14}
$$

The scalar curvature (4) therefore takes the following form:

$$
R=\frac{2\dot{q}}{Nz}+\frac{6k}{z}\equiv R_q=\frac{2}{Nz}(v+3Nk),
\tag{15}
$$

Equation (10) can be expressed as

$$
A_{1q}=\int\left[2\alpha\sqrt{z}(v+3kN)+\frac{4\beta}{N\sqrt{z}}(v+3kN)^2+z^{\frac{3}{2}}\left(\frac{v_\phi^2}{2N}-NV\right)\right]dt,
\tag{16}
$$

while the Lagrangian density is

$$
L_{1q}=2\alpha\sqrt{z}(v+3kN)+\frac{4\beta}{N\sqrt{z}}(v+3kN)^2+z^{\frac{3}{2}}\left(\frac{v_\phi^2}{2N}-NV\right).
\tag{17}
$$

Note that the boundary terms remain intact in the equation, as well as in the point Lagrangian. Canonical momenta are

$$
p_q=\frac{\partial L_q}{\partial v}=2\alpha\sqrt{z}+\frac{8\beta}{N\sqrt{z}}(v+3kN),\ p_N=\frac{\partial L_{1q}}{\partial v_N}=0,\ p_z=\frac{\partial L_q}{\partial v_z}=0\ \text{and}\ p_\phi=\frac{\partial L_q}{\partial v_\phi}=\frac{z^{\frac{3}{2}}v_\phi}{N}.
\tag{18}
$$

Clearly, there exist two primary constraints, $C \equiv p_N \approx 0$, and $D \equiv p_z \approx 0$. Therefore, Dirac constraint analysis appears to be essential. However, there is a wonderful twist in the BL formalism. For example, one can express the modified Lagrangian density as

$$L_1^* = L_{1q} + p_q(\dot{q} - v) + p_N(\dot{N} - v_N) + p_z\left(\dot{z} - \frac{2Nq}{3}\right) + p_\phi(\dot{\phi} - v_\phi), \tag{19}$$

and, equivalently, the Hamiltonian density as

$$H_1^* = p_q\dot{q} + p_N\dot{N} + p_\phi\dot{\phi} + p_z\dot{z} - L_1^* = p_q v + p_N v_N + p_\phi v_\phi + p_z\frac{2Nq}{3} - L_{1q}. \tag{20}$$

As a consequence, one can immediately find that the primary constraint $D \equiv p_z \approx 0$ disappears. Furthermore, since $N$ is a non-dynamical Lagrange multiplier, the constraint $C$ vanishes strongly. Therefore, one arrives at

$$H_1^* = p_q v + C v_N + p_\phi v_\phi + p_z\frac{2qN}{3} - L_{1q} = C v_N + p_q v + p_\phi v_\phi + p_z\frac{2qN}{3} - L_{1q} = C v_N + N\mathcal{H}_{BL}^m, \tag{21}$$

where

$$N\mathcal{H}_{BL}^m = p_q v + p_\phi v_\phi + \frac{2N}{3}q p_z - L_{1q}$$

$$= p_q v + p_\phi v_\phi + \frac{2N}{3}q p_z - 2\alpha\sqrt{z}(v + 3kN) - \frac{4\beta}{N\sqrt{z}}(v + 3kN)^2 - z^{\frac{3}{2}}\left(\frac{v_\phi^2}{2N} + NV\right). \tag{22}$$

In the above, $m$ in the superscript stands for minimally coupled theory. Upon substituting $v$ and $v_\phi$ from the definition of momentum (18), we obtain,

$$N\mathcal{H}_{BL}^m = N\left[\frac{2q}{3}p_z + \frac{\sqrt{z}}{16\beta}p_q^2 - \left(\frac{\alpha z}{4\beta} + 3k\right)p_q + \frac{\alpha^2}{4\beta}z^{\frac{3}{2}} + \frac{1}{2z^{\frac{3}{2}}}p_\phi^2 + Vz^{\frac{3}{2}}\right], \tag{23}$$

so that the canonical Hamiltonian finally reads as

$$\mathcal{H}_{BL}^m = \frac{2q}{3}p_z + \frac{\sqrt{z}}{16\beta}p_q^2 - \left(\frac{\alpha z}{4\beta} + 3k\right)p_q + \frac{\alpha^2}{4\beta}z^{\frac{3}{2}} + \frac{1}{2z^{\frac{3}{2}}}p_\phi^2 + Vz^{\frac{3}{2}}. \tag{24}$$

The action (10) may also be cast in the canonical form as,

$$A_{1q} = \int\left(\dot{z}p_z + \dot{q}p_q + \dot{\phi}p_\phi - N\mathcal{H}_{BL}\right)dt\,d^3x \;=\; \int\left(\dot{h}_{ij}\pi^{ij} + \dot{K}_{ij}\Pi^{ij} + \dot{\phi}p_\phi - N\mathcal{H}_{BL}\right)dt\,d^3x, \tag{25}$$

where $\pi^{ij}$ and $\Pi^{ij}$ are momenta canonically conjugate to $h_{ij}$ and $K_{ij}$, respectively. For the sake of comparison, let us make the following canonical transformation:

$$q \to \frac{3}{2}x; \quad p_q \to \frac{2}{3}p_x, \tag{26}$$

to express the above Hamiltonian (24) as

$$\mathcal{H}_{BL}^m = xp_z + \frac{\sqrt{z}}{36\beta}p_x^2 - \left(\frac{\alpha z}{6\beta} + 2k\right)p_x + \frac{\alpha^2 z^{\frac{3}{2}}}{4\beta} + \frac{p_\phi^2}{2z^{\frac{3}{2}}} + Vz^{\frac{3}{2}}. \tag{27}$$

It is revealed that the above Hamiltonian (27) is exactly the one obtained earlier, following Ostrogradsky's, Dirac's, and Horowitz's formalisms [15]. Note that here, very much like Ostrogradsky's and Dirac's formalisms, once the formalism is initiated, i.e., $R$ is expressed in terms of $\{h_{ij}, K_{ij}\}$ (15) as well as the action (16) and the point Lagrangian (17), there remains no option to integrate the action by parts. As a result, even the Gibbons–Hawking–York term [20,21], which is physically meaningful, being associated with the entropy of the black hole, along with its higher-order counterpart, remains obscure. On the contrary,

following the modified Horowitz formalism (MHF), where boundary terms are taken care of, we earlier obtained [15]

$$\mathcal{H}^m_{MHF} = xp_z + \frac{\sqrt{z}}{36\beta}p_x^2 + \frac{3\alpha x^2}{2\sqrt{z}} - \frac{18\beta kx^2}{z^{\frac{3}{2}}} - \frac{36\beta k^2}{\sqrt{z}} - 6k\alpha\sqrt{z} + \frac{p_\phi^2}{2z^{\frac{3}{2}}} + Vz^{\frac{3}{2}}. \tag{28}$$

Although (27) and (28) exactly match under the following set of canonical transformations,

$$p_z \to p_z - \frac{18k\beta x}{z^{\frac{3}{2}}} + \frac{3\alpha x}{2\sqrt{z}}, \quad z \to z,$$

$$p_x \to p_x + \frac{36k\beta}{\sqrt{z}} - 3\alpha\sqrt{z}, \quad x \to x,$$

$$p_\phi \to p_\phi, \quad \phi \to \phi.$$

and apparently there is no contradiction between the two, note the essential difference: the linear term in the momentum ($p_x$), which is very much present in (27), remains absent from the Hamiltonian (28). As a result, the two Hamiltonians (27) and (28) induce completely different quantum dynamics, since in the quantum domain, as previously mentioned, canonical transformation cannot be carried over due to non-linearity.

*2.2. Non-Minimally Coupled Case*

We find that the two different Hamiltonians (27) and (28) render two different quantum descriptions of the same classical model. Although some of the essential features (Gibbons–Hawking–York term and its higher-order counterpart) are absent from the Hamiltonian (27), it is not clear which one gives the correct quantum description of the theory. Furthermore, there may exist a unitary transformation (which we have not yet found) relating the two Hamiltonian operators. Therefore, to realize the situation more deeply, we consider the non-minimally coupled case next, whose action

$$A_2 = \int \sqrt{-g}\, d^4x \left[ f(\phi)R + \beta R^2 - \frac{1}{2}\phi_{,\mu}\phi^{,\nu} - V(\phi) \right] + f(\phi)\Sigma_R + B\Sigma_{R^2}, \tag{29}$$

may be expressed in the RW metric (1) as

$$A_2 = \int \left[ 3f(\phi)\sqrt{z}\left(\frac{\ddot{z}}{N} - \frac{\dot{N}\dot{z}}{N^2} + 2kN\right) + \frac{9\beta}{\sqrt{z}}\left(\frac{\ddot{z}^2}{N^3} - \frac{2\dot{N}\dot{z}\ddot{z}}{N^4} + \frac{\dot{N}^2\dot{z}^2}{N^5} - \frac{4k\dot{N}\dot{z}}{N^2} + \frac{4k\ddot{z}}{N} + 4k^2N\right) \right.$$
$$\left. + z^{\frac{3}{2}}\left(\frac{\dot{\phi}^2}{2N} - VN\right) \right] dt + f(\phi)\Sigma_R + B\Sigma_{R^2}, \tag{30}$$

where, as already mentioned, the supplementary boundary terms are required when MHF is taken into account. In the above, we consider an arbitrary functional coupling parameter $f(\phi)$. Pursuing the same procedure as above, one finally arrives at the following Hamiltonian:

$$\mathcal{H}^{nm}_{BL} = xp_z + \frac{\sqrt{z}}{36\beta}p_x^2 - \left(f(\phi)\frac{z}{6\beta} + 2k\right)p_x + f^2(\phi)\frac{z^{\frac{3}{2}}}{4\beta} + \frac{p_\phi^2}{2z^{\frac{3}{2}}} + Vz^{\frac{3}{2}}, \tag{31}$$

which is again identical to the one found following Dirac's formalism and may be found following Ostrogradsky's and Horowitz's techniques as well [22]. In the superscript, *nm* stands for non-minimal coupling. The action (30) may also be cast in the canonical form as in (25). On the contrary, following MHF, one finds [22]

$$\mathcal{H}^{nm}_{MHF} = xp_z + \frac{\sqrt{z}}{36\beta}p_x^2 + 3f(\phi)\left(\frac{x^2}{2\sqrt{z}} - 2k\sqrt{z}\right) - \frac{18k\beta}{\sqrt{z}}\left(\frac{x^2}{z} + 2k\right) + \frac{p_\phi^2}{2z^{\frac{3}{2}}}$$
$$+ \frac{3xf'(\phi)p_\phi}{z} + \frac{9f'(\phi)^2x^2}{2\sqrt{z}} + Vz^{\frac{3}{2}}. \tag{32}$$

However, under the following set of canonical transformations,

$$p_z \to p_z - \frac{18\beta kx}{z^{\frac{3}{2}}} + \frac{3f(\phi)x}{2\sqrt{z}}, \ z \to z,$$

$$p_x \to p_x + 36\beta \frac{k}{\sqrt{z}} + 3f(\phi)\sqrt{z}, \ x \to x,$$

$$p_\phi \to p_\phi + 3f'(\phi)x\sqrt{z}, \ \phi \to \phi,$$

the two Hamiltonians (31) and (32) match again [22]. Nevertheless, here the difference is predominant and explicit. Note that the $f'(\phi)$ term does not appear in (31), while it is coupled with $p_\phi$ in (32). This coupled ($f'(\phi)p_\phi$) term requires operator ordering in the quantum domain, which is different for different forms of $f(\phi)$. Hence, even if the two Hamiltonians are related through unitary transformation, such a transformation would be different for different forms of $f(\phi)$.

### 2.3. Einstein–Gauss–Bonnet–Dilatonic Action in the Presence of Higher-Order Terms

Although it is clear that two different quantum descriptions follow from the same classical action using different techniques, it is still abstruse to select the correct description. Therefore, we next consider Einstein–Gauss–Bonnet–dilatonic coupled action in the presence of a higher-order curvature-invariant term. The Gauss–Bonnet (GB) term arises quite naturally as the leading order of the $\alpha'$ expansion of heterotic superstring theory, where $\alpha'$ is the inverse string tension [41–46]. Several interesting features of the GB term have been explored in the literature [47–67]. However, the Gauss–Bonnet term is topological-invariant in four dimensions, and so to measure its contribution to the field equations, dynamic dilatonic scalar coupling is required. It is worth mentioning that, in string-induced gravity near initial singularity, GB coupling with a scalar field plays a crucial role in the occurrence of nonsingular cosmology [68,69]. A particular hallmark of the GB term is the fact that, despite being formed from a combination of higher-order curvature-invariant terms ($\mathcal{G} = R^2 - 4R_{\mu\nu}R^{\mu\nu} + R_{\alpha\beta\mu\nu}R^{\alpha\beta\mu\nu}$) (3), it ends up only with second-order field equations, avoiding Ostrogradsky's instability, and equivalently, ghost degrees of freedom. Nonetheless, such a wonderful feature ultimately leads to a serious pathology, the 'Branched Hamiltonian', which has no unique resolution to date [70–72]. Nevertheless, it has been revealed that, by supplementing the action with a higher-order curvature-invariant term, the pathology may be bypassed [24,25]. We therefore consider the following action:

$$A_3 = \int \sqrt{-g}\, d^4x \left[ \alpha R + \beta R^2 + \gamma(\phi)\mathcal{G} - \frac{1}{2}\phi_{,\mu}\phi^{,\nu} - V(\phi) \right] + \alpha\Sigma_R + \beta\Sigma_{R^2} + \gamma(\phi)\Sigma_{\mathcal{G}}. \quad (33)$$

In the above, the Gauss–Bonnet term $\mathcal{G}$ is coupled with $\gamma(\phi)$, while $V(\phi)$ is the dilatonic potential. Furthermore, the symbol $\mathcal{K}$ stands for $\mathcal{K} = K^3 - 3KK^{ij}K_{ij} + 2K^{ij}K_{ik}K_j^k$, where $K$ is the trace of the extrinsic curvature tensor $K_{ij}$, and $\gamma(\phi)\Sigma_{\mathcal{G}} = 4\gamma(\phi) \oint_{\partial\mathcal{V}} \left( 2G_{ij}K^{ij} + \frac{\mathcal{K}}{3} \right) \sqrt{h}d^3x$ is the supplementary boundary term associated with the Gauss–Bonnet sector. The $\binom{0}{0}$ component of Einstein's field equation in terms of the scale factor here reads as

$$\frac{6\alpha}{a^2}\left( \frac{\dot{a}^2}{N^2} + k \right) + \frac{36\beta}{a^2N^4}\left( 2\dot{a}\,\dddot{a} - 2\dot{a}^2\frac{\ddot{N}}{N} - \ddot{a}^2 - 4\dot{a}\ddot{a}\frac{\dot{N}}{N} + 2\dot{a}^2\frac{\ddot{a}}{a} + 5\dot{a}^2\frac{\dot{N}^2}{N^2} - 2\frac{\dot{a}^3\dot{N}}{aN} \right.$$

$$\left. - 3\frac{\dot{a}^4}{a^2} - 2kN^2\frac{\dot{a}^2}{a^2} + \frac{k^2N^4}{a^2} \right) + \frac{24\gamma'\dot{a}\dot{\phi}}{N^2a^3}\left( \frac{\dot{a}^2}{N^2} + k \right) - \left( \frac{\dot{\phi}^2}{2N^2} + V \right) = 0. \quad (34)$$

Equation (33) in terms of the basic variable ($h_{ij} = a^2\delta_{ij} = z\delta_{ij}$) may be expressed as

$$A_3 = \int \left[ 3\alpha\sqrt{z}\left( \frac{\ddot{z}}{N} - \frac{\dot{N}\dot{z}}{N^2} + 2kN \right) + \frac{9\beta}{\sqrt{z}}\left( \frac{\dot{z}^2}{N^3} - \frac{2\dot{N}\dot{z}\ddot{z}}{N^4} + \frac{\dot{N}^2\dot{z}^2}{N^5} - \frac{4k\dot{N}\dot{z}}{N^2} + \frac{4k\ddot{z}}{N} + 4k^2 N \right) \right.$$

$$\left. + \frac{3\gamma(\phi)}{N\sqrt{z}}\left( \frac{\dot{z}^2\ddot{z}}{N^2 z} + 4k\ddot{z} - \frac{\dot{z}^4}{2N^2 z^2} - \frac{\dot{N}\dot{z}^3}{N^3 z} - \frac{2k\dot{z}^2}{z} - \frac{4k\dot{N}\dot{z}}{N} \right) + z^{\frac{3}{2}}\left( \frac{1}{2N}\dot{\phi}^2 - VN \right) \right] dt \quad (35)$$

$$+ \alpha\Sigma_R + \beta\Sigma_{R^2} + \gamma(\phi)\Sigma_{\mathcal{G}},$$

where the additional supplementary boundary term $\gamma(\phi)\Sigma_{\mathcal{G}} = -\gamma(\phi)\frac{\dot{z}}{N\sqrt{z}}\left( \frac{\dot{z}^2}{N^2 z} + 12k \right)$ is required in the case of MHF. Inserting the other basic variable ($K_{ij} = -\frac{q}{3}\delta_{ij}$) and considering $\dot{q} = v$ (13), Equation (35) finally may be expressed as

$$A_{3q} = \int \left[ 2\alpha\sqrt{z}(v + 3kN) + \frac{4\beta}{N\sqrt{z}}(v + 3kN)^2 + \frac{8\gamma(\phi)}{\sqrt{z}}\left[ \frac{vq^2}{9z} - \frac{q^4 N}{27z^2} + kv - \frac{kNq^2}{3z} \right] \right.$$

$$\left. + z^{\frac{3}{2}}\left( \frac{v_\phi^2}{2N} - NV \right) \right] dt + \alpha\Sigma_R + \beta\Sigma_{R^2} + \Lambda(\phi)\Sigma_{\mathcal{G}}. \quad (36)$$

Thus, the Lagrangian density takes the following form,

$$L_{3q} = 2\alpha\sqrt{z}(v + 3kN) + \frac{4\beta}{N\sqrt{z}}(v + 3kN)^2 + \frac{8\gamma(\phi)}{\sqrt{z}}\left[ \frac{vq^2}{9z} - \frac{q^4 N}{27z^2} + kv - \frac{kNq^2}{3z} \right] + z^{\frac{3}{2}}\left( \frac{1}{2N}v_\phi^2 - VN \right), \quad (37)$$

where boundary terms are not taken care of. The canonical momenta are

$$p_q = \frac{\partial L_q}{\partial v} = 2\alpha\sqrt{z} + \frac{8\beta}{N\sqrt{z}}(v + 3kN) + \frac{8\gamma(\phi)}{\sqrt{z}}\left( \frac{q^2}{9z} + k \right),$$

$$p_N = \frac{\partial L_{3q}}{\partial v_N} = 0, \quad p_\phi = \frac{\partial L_q}{\partial v_\phi} = \frac{z^{\frac{3}{2}} v_\phi}{N}, \quad \text{and,} \quad p_z = \frac{\partial L_{3q}}{\partial v_z} = 0. \quad (38)$$

Clearly, there exist two primary constraints, $C \equiv p_N \approx 0$ and $D \equiv p_z \approx 0$, which are usually handled by Dirac constraint analysis. However, as mentioned, such analysis is not at all required in the BL formalism. For example, one can express the modified Lagrangian density as

$$L_3^* = L_{3q} + p_q(\dot{q} - v) + p_N(\dot{N} - v_N) + p_z\left( \dot{z} - \frac{2Nq}{3} \right) + p_\phi(\dot{\phi} - v_\phi), \quad (39)$$

so that the corresponding Hamiltonian density takes the following form:

$$H_3^* = p_q\dot{q} + p_N\dot{N} + p_\phi\dot{\phi} + p_z\dot{z} - L_3^* = p_q v + p_N v_N + p_\phi v_\phi + \frac{2Nq}{3}p_z - L_{3q}. \quad (40)$$

In the process, the primary constraint $D \equiv p_z \approx 0$ disappears, and one obtains

$$H_3^* = p_q v + C v_N + p_\phi v_\phi + p_z\frac{2qN}{3} - L_{3q} = C v_N + \left( p_q v + p_\phi v_\phi + \frac{2qN}{3}p_z - L_{3q} \right) = C v_N + N\mathcal{H}^{GB}{}_{BL}. \quad (41)$$

In the superscript, $GB$ represents the Hamiltonian for Einstein–Gauss–Bonnet–dilatonic coupling. Note that the constraint $C \equiv p_N$ strongly vanishes, since the lapse function $N$ is simply a Lagrange multiplier. Therefore,

$$N\mathcal{H}^{GB}{}_{BL} = p_q v + p_\phi v_\phi + \frac{2qN}{3}p_z - L_{3q}$$

$$= p_q v + p_\phi v_\phi + \frac{2qN}{3}p_z - 2\alpha\sqrt{z}(v + 3kN) - \frac{4\beta}{N\sqrt{z}}(v + 3kN)^2 \quad (42)$$

$$- \frac{8\gamma(\phi)}{\sqrt{z}}\left[ \frac{vq^2}{9z} - \frac{q^4 N}{27z^2} + kv - \frac{kNq^2}{3z} \right] - z^{\frac{3}{2}}\left( \frac{1}{2N}v_\phi^2 - VN \right).$$

Upon substituting $v$ from the definition of momentum (38), one obtains

$$
N\mathcal{H}^{GB}{}_{BL} = N\left[\frac{2qp_z}{3} + \frac{\sqrt{z}p_q^2}{16\beta} - p_q\left(\frac{\alpha z}{4\beta} + 3k\right) + \frac{\alpha^2 z^{\frac{3}{2}}}{4\beta} - p_q\left(\frac{\gamma q^2}{9\beta z} + \frac{\gamma k}{\beta}\right) + \frac{2\alpha\gamma}{\beta}\left(\frac{q^2}{9\sqrt{z}} + k\sqrt{z}\right) \right.
$$
$$
\left. + \frac{4\gamma q^4}{27z^{\frac{3}{2}}}\left(\frac{\gamma}{3\beta} + 2\right) + \frac{8\gamma k q^2}{3z^{\frac{3}{2}}}\left(\frac{\gamma}{3\beta} + 2\right) + \frac{12\gamma k^2}{\sqrt{z}}\left(\frac{\gamma}{3\beta} + 2\right) + \frac{p_\phi^2}{2z^{\frac{3}{2}}} + Vz^{\frac{3}{2}}\right].
\tag{43}
$$

The canonical Hamiltonian therefore finally reads as

$$
\mathcal{H}^{GB}{}_{BL} = \frac{2qp_z}{3} + \frac{\sqrt{z}p_q^2}{16\beta} - p_q\left(\frac{\alpha z}{4\beta} + 3k\right) + \frac{\alpha^2 z^{\frac{3}{2}}}{4\beta} - p_q\left(\frac{\gamma q^2}{9\beta z} + \frac{\gamma k}{\beta}\right) + \frac{2\alpha\gamma}{\beta}\left(\frac{q^2}{9\sqrt{z}} + k\sqrt{z}\right)
$$
$$
+ \frac{4\gamma q^4}{27z^{\frac{3}{2}}}\left(\frac{\gamma}{3\beta} + 2\right) + \frac{8\gamma k q^2}{3z^{\frac{3}{2}}}\left(\frac{\gamma}{3\beta} + 2\right) + \frac{12\gamma k^2}{\sqrt{z}}\left(\frac{\gamma}{3\beta} + 2\right) + \frac{p_\phi^2}{2z^{\frac{3}{2}}} + Vz^{\frac{3}{2}}.
\tag{44}
$$

Again, for the sake of comparison, let us use the canonical transformation $q \to \frac{3}{2}x$; $p_q \to \frac{2}{3}p_x$ (26) to express the above Hamiltonian (44) in the following form:

$$
\mathcal{H}^{GB}{}_{BL} = xp_z + \frac{\sqrt{z}p_x^2}{36\beta} + \frac{\alpha^2 z^{\frac{3}{2}}}{4\beta} - \left(\frac{\alpha z}{6\beta} + \frac{\gamma x^2}{6\beta z} + \frac{2k\gamma}{3\beta} + 2k\right)p_x + \frac{p_\phi^2}{2z^{\frac{3}{2}}} + \left(\frac{\gamma^2}{4\beta z^{\frac{5}{2}}} + \frac{3\gamma}{2z^{\frac{5}{2}}}\right)x^4
$$
$$
+ \left(\frac{\alpha\gamma}{2\beta\sqrt{z}} + \frac{12k\gamma}{z^{\frac{3}{2}}} + \frac{2k\gamma^2}{\beta z^{\frac{3}{2}}}\right)x^2 + \frac{2\alpha k\gamma\sqrt{z}}{\beta} + \frac{24k^2\gamma}{\sqrt{z}} + \frac{4k^2\gamma^2}{\beta\sqrt{z}} + Vz^{\frac{3}{2}},
\tag{45}
$$

Note that it is similar to the one already found following Dirac's formalism, and may be found following Ostrogradsky's and Horowitz's techniques as well [22]. The action (35) may also be cast in the canonical form with respect to the basic variables as

$$
A_{3q} = \int\left(\dot{z}p_z + \dot{q}p_q + \dot{\phi}v_\phi - N\mathcal{H}_{BL}\right)dt\,d^3x = \int\left(\dot{h}_{ij}\pi^{ij} + \dot{K}_{ij}\Pi^{ij} + \dot{\phi}v_\phi - N\mathcal{H}_{MHF}\right)dt\,d^3x, \tag{46}
$$

where $\pi^{ij}$ and $\Pi^{ij}$ are momenta canonically conjugate to $h_{ij}$ and $K_{ij}$, respectively. Hence, everything appears to be consistent. On the contrary, following MHF, one finds [22]

$$
\mathcal{H}^{GB}{}_{MHF} = xp_z + \frac{\sqrt{z}p_x^2}{36\beta} + 3\alpha\left(\frac{x^2}{2\sqrt{z}} - 2k\sqrt{z}\right) - \frac{18k\beta}{\sqrt{z}}\left(\frac{x^2}{z} + 2k\right) + \left(\frac{x^6}{2z^{\frac{9}{2}}} + \frac{12kx^4}{z^{\frac{7}{2}}} + \frac{72k^2x^2}{z^{\frac{5}{2}}}\right)\gamma'^2
$$
$$
+ \left(\frac{x^3}{z^3} + \frac{12kx}{z^2}\right)\gamma'p_\phi + \frac{p_\phi^2}{2Z^{\frac{3}{2}}} + Vz^{\frac{3}{2}}.
\tag{47}
$$

Nonetheless, under the following set of canonical transformations,

$$
p_z \to p_z - \frac{18\beta kx}{z^{\frac{3}{2}}} + \frac{3\alpha x}{2\sqrt{z}} - \frac{6k\gamma(\phi)x}{z^{\frac{3}{2}}} - \frac{3\gamma(\phi)x^3}{2z^{\frac{5}{2}}},\ \ z \to z,
$$
$$
p_x \to p_x + 36\beta\frac{k}{\sqrt{z}} + 3\alpha\sqrt{z} + \frac{3\gamma(\phi)x^2}{z^{\frac{3}{2}}} + \frac{12k\Lambda}{\sqrt{z}},\ \ x \to x,
\tag{48}
$$
$$
p_\phi \to p_\phi - \frac{\gamma'(\phi)x^3}{z^{\frac{3}{2}}} - \frac{12k\gamma'(\phi)x}{\sqrt{z}},\ \ \phi \to \phi,
$$

the two Hamiltonians (45) and (47) match again [22]. Therefore, it appears there is no problem. Nevertheless, note that the Hamiltonian (47) contains the term $(\gamma'(\phi)p_\phi)$, which is absent from (45). Now, during canonical quantization, the presence of this term requires operator ordering, which is different for different forms of $\gamma(\phi)$. As a result, even if the two may be related through unitary transformation, such a transformation would be different for different forms of $\gamma(\phi)$. Thus, there does not exist a unique unitary transformation. In a nutshell, we find that the two Hamiltonians (45) and (47) induce two different descriptions in the quantum domain, and there is apparently no way to choose which one is correct.

### 3. The Role of Divergent Terms

The first important point to mention is that in all the formalisms, the scale factor is treated as the basic variable, while we initiate our program using the three-space curvature instead. To explain the reason behind this choice, let us consider curvature squared action, $A = \int \beta R^2 d^4 x$, as an example. Under variation, it gives a total derivative term $\sigma = -4\beta \int RK\sqrt{h}\, d^3 x$, as mentioned earlier, where $K$ is the trace of the extrinsic curvature tensor $K_{ij}$. A counter term $(-\sigma)$, known as the modified Gibbons–Hawking–York term [20,21], must be added to the action in case, instead of $\delta \dot{q}$, $\delta R$ is kept fixed at the boundary, as in MHF. In the RW (1) metric under consideration, the action reads as

$$A = 36\beta \int \left[ a\ddot{a}^2 + 2\dot{a}^2\ddot{a} + 2k\ddot{a} + \frac{\dot{a}^4}{a} + \frac{2k\dot{a}^2}{a} + \frac{k^2}{a} \right] dt \int d^3 x. \tag{49}$$

Under integration by parts, we end up with

$$A = C \int \left[ a\ddot{a}^2 + \frac{\dot{a}^4}{a} + \frac{2k\dot{a}^2}{a} + \frac{k^2}{a} \right] dt + C\left( \frac{2}{3}\dot{a}^3 + 2k\dot{a} \right). \tag{50}$$

where $C = 36\beta \int d^3 x$. Following Horowitz's program, we introduce an auxiliary variable $Q = \frac{\partial A}{\partial \ddot{a}} = 2Ca\ddot{a}$ into the action in the following manner, such that it may be cast in canonical form:

$$A = \int \left[ Q\ddot{a} - \frac{Q^2}{4Ca} + C\left( \frac{\dot{a}^4}{a} + \frac{2k\dot{a}^2}{a} + \frac{k^2}{a} \right) \right] dt + C\left( \frac{2}{3}\dot{a}^3 + 2k\dot{a} \right). \tag{51}$$

Integrating the action again by parts, we find

$$A = \left[ -\dot{Q}\dot{a} - \frac{Q^2}{4Ca} + C\left( \frac{\dot{a}^4}{a} + \frac{2k\dot{a}^2}{a} + \frac{k^2}{a} \right) \right] + C\left( \frac{Q\dot{a}}{C} + \frac{2}{3}\dot{a}^3 + 2k\dot{a} \right). \tag{52}$$

The action is canonical, since the Hessian determinant is non-zero. It is trivial to check that the above action gives correct field equations, but the left-out total derivative term may be expressed as

$$\sigma' = -4\beta \int RK\sqrt{h}\, d^3 x + 16\beta \int K\sqrt{h}\left( \frac{\dot{a}^2}{a^2} \right) d^3 x. \tag{53}$$

As a result, $\sigma \neq \sigma'$. Thus, some redundant total derivative terms are pulled out in the process, which has severe consequences in the quantum domain. On the contrary, if we start with $z = a^2$, the action reads as

$$A = C \int \left( \frac{\ddot{z}^2}{4\sqrt{z}} + \frac{k\ddot{z}}{\sqrt{z}} + \frac{k^2}{\sqrt{z}} \right) dt = C \int \left( \frac{\ddot{z}^2}{4\sqrt{z}} + \frac{2}{\sqrt{z}} \right) + C\frac{k\dot{z}}{\sqrt{z}}, \tag{54}$$

where the last expression is found under integration by parts. Then, following Horowitz's program, we find the auxiliary variable $Q = \frac{\partial A}{\partial \ddot{z}} = C\frac{\ddot{z}}{2\sqrt{z}}$, and by judiciously introducing it in the action we find

$$A = \int \left[ Q\ddot{z} - \frac{\sqrt{z}Q^2}{C} + C\left( \frac{k\ddot{z}}{\sqrt{z}} + \frac{k^2}{\sqrt{z}} \right) \right] dt + C\frac{k\dot{z}}{\sqrt{z}}, \tag{55}$$

Finally, performing integration by parts again, one obtains

$$A = \int \left[ -\dot{Q}\dot{z} - \frac{\sqrt{z}Q^2}{C} + C\left( \frac{k\ddot{z}}{\sqrt{z}} + \frac{k^2}{\sqrt{z}} \right) \right] dt + C\left[ \frac{Q\dot{z}}{C} + \frac{k\dot{z}}{\sqrt{z}} \right], \tag{56}$$

The action is again canonical, and the Euler–Lagrange equations here again lead to the appropriate field equations, while one can express the total derivative term as $\sigma$. In a nutshell, although total derivative terms do not affect the classical field equations, for non-linear theories, such as gravity, such terms have an effect on the quantum dynamics.

Therefore, to establish consistence in every respect, $h_{ij}$ should be treated as the basic variable, instead of the scale factor. This is essentially the so-called MHF, which finally requires the auxiliary variable to be replaced by the the second basic variable, viz. the extrinsic curvature tensor $K_{ij} = -a\dot{a} = -\dot{z} = x(\text{say})$, in the Hamiltonian.

Next, we observe that the phase-space structures obtained following BL formalism, although are identical to the Ostrogradsky/Dirac/Horowitz formalism, all differ from the MHF up to a canonical transformation. We quote from [22] the general argument in connection with the total derivative terms: "it is just the change of the variables in the wave function and the phase transformation, plus the change of the integration measure, and the transformation of the momenta respecting the change of the measure, and so a unitary transformation relates the two". It is possible (although we have not found this) that each pair of quantum equations cast from (27) and (28), (31) and (32), and (45) and (47) are related by unitary transformation. However, it was also mentioned [22] that different forms of coupling parameter yield different quantum dynamics in the case of MHF, due to the presence of a coupling term $(f'(\phi)p_\phi)$ for the non-minimally coupled case, and $(\gamma'(\phi)p_\phi)$ for the Einstein–Gauss–Bonnet–dilaton coupled case, in the Hamiltonian. Thus, different unitary transformations (if they exist) are required to relate the last two pairs. Such coupling, as well as the derivative of the coupling parameter, remain absent in other formalisms. In a nutshell, the unitary transformation relating each pair is not unique. Furthermore, the semiclassical wave functions found for all three cases studied here exhibit different pre-factors and exponents for each pair [22]. This generates a different probability amplitude and evolution of the wave function while entering the classical domain.

Finally, it is important to note that, if the coupling parameter $f(\phi)$ is treated as constant in case of Section 2.2 , the Hamiltonian (32) merely reduces to (28), while the Hamiltonian (31) reduces to (27). Hence, the question is which of the two should be treated as the correct quantum description of the models under consideration? In this connection, we mention that a serious problem arises with Ostrogradsky/Dirac/Horowitz and BL formalisms when considering Gauss–Bonnet–dilaton-induced action. To be specific, in Section 2.3 if $\gamma(\phi)$ is treated as a constant, then the contribution of the Gauss–Bonnet term disappears from the Hamiltonian (47), and reduces to (28). Indeed it should be, since, as mentioned, the Gauss–Bonnet term is topologically invariant in four dimensions, and so without functional coupling, it should not contribute to the field equations and the Hamiltonian as well. On the contrary, a constant $\gamma$ does not affect the form of the Hamiltonian (45), and it does not reduce to (27). This means that if we had started with a constant $\gamma$ from the beginning, all the terms appearing with $\gamma$ in (45) would have been absent, and the end result would be (27). After constructing the Hamiltonian with arbitrary $\gamma = \gamma(\phi)$, if we set it equal to a constant, then its contribution remains present, and we obtain a different Hamiltonian altogether. Clearly, this is wrong. Hence, we realize that boundary terms indeed play a crucial role while constructing the phase-space structure of non-linear theories. In fact, if boundary terms are taken into account from the very beginning, treating $h_{ij}$ as the basic variable, then Horowitz's formalism reduces to the MHF, as already demonstrated. It was also noticed that if the Dirac algorithm is applied after integrating the action by parts, then it also yields a Hamiltonian identical to MHF [22]. It is therefore suggested to conduct the same test for the BL formalism too. In this section, we shall first integrate actions by parts to remove the total derivative terms and follow the BL formalism thereafter, in order to explore the outcome.

### 3.1. Scalar–Tensor Theory: Minimal Coupling

Upon integrating Equation (30) by parts, we obtain

$$A_1 = \int \left[ -\frac{3\alpha \dot{z}^2}{2N\sqrt{z}} + 6\alpha kN\sqrt{z} + \frac{9\beta}{N\sqrt{z}} \left\{ \left( \frac{\ddot{z}}{N} - \frac{\dot{N}\dot{z}}{N^2} \right)^2 + \frac{2k\dot{z}^2}{z} + 4k^2 N^2 \right\} + z^{\frac{3}{2}} \left( \frac{\dot{\phi}^2}{2N} - NV \right) \right] dt. \quad (57)$$

Replacing $\dot{z}$ by $\frac{2N}{3}q$ in view of (13), the above action may be cast as

$$A_{1q} = \int \left[ -\frac{2}{3}\alpha N \frac{q^2}{\sqrt{z}} + 6\alpha k N \sqrt{z} + \frac{9\beta}{N\sqrt{z}} \left( \frac{4}{9}\dot{q}^2 + \frac{8kN^2q^2}{9z} + 4k^2N^2 \right) + z^{\frac{3}{2}} \left( \frac{\dot{\phi}^2}{2N} - NV \right) \right] dt. \quad (58)$$

Note that the equation cannot be expressed only in terms of velocities, due to the explicit presence of $q$, unlike (16). However, a similar situation is arrived at in the Einstein–Gauss–Bonnet–dilaton case, and so it does not matter. The canonical momenta are as follows:

$$p_q = \frac{8\beta}{N\sqrt{z}}\dot{q}; \quad p_\phi = \frac{z^{\frac{3}{2}}}{N}\dot{\phi}; \quad p_z = 0 = p_N. \quad (59)$$

Dirac constraint analysis appears to be inevitable, since the action is singular. However, as mentioned, for the lapse function $N$, being the Lagrange multiplier, the constraint strongly vanishes, so that one can ignore it without the loss of generality. Still, another primary constraint $p_z = 0$ is apparent. Nonetheless, as already mentioned, in the BL formalism, Dirac analysis may be bypassed despite the presence of the constraint $p_z = 0$ in the following manner, expressing the Lagrangian density as,

$$L_{1q} = -\frac{2}{3}\alpha N \frac{q^2}{\sqrt{z}} + 6\alpha k N \sqrt{z} + \frac{9\beta}{N\sqrt{z}} \left( \frac{4}{9}\dot{q}^2 + \frac{8kN^2q^2}{9z} + 4k^2N^2 \right) + z^{\frac{3}{2}} \left( \frac{\dot{\phi}^2}{2N} - NV \right), \quad (60)$$

and the Hamiltonian reads as

$$\begin{aligned} NH^m_{MBL} &= p_q\dot{q} + p_z\dot{z} + p_\phi\dot{\phi} - L_{1q} \\ &= \frac{N\sqrt{z}}{16\beta}p_q^2 + \frac{2}{3}Nqp_z + \frac{N}{2z^{\frac{3}{2}}}p_\phi^2 + \frac{2\alpha Nq^2}{3\sqrt{z}} - 6\alpha k N \sqrt{z} - \frac{8\beta k Nq^2}{z^{\frac{3}{2}}} - \frac{36\beta k^2 N}{\sqrt{z}} + NVz^{\frac{3}{2}}, \end{aligned} \quad (61)$$

where we have used (59) and replaced $\dot{z}$ by $\frac{2N}{3}q$, in view of (13). The suffix {MBL} now represents the modified Buchbinder–Lyakhovich formalism. Finally, as before, for the sake of comparison, if we perform the canonical transformation $q \to \frac{3}{2}x$, and $p_q \to \frac{2}{3}p_x$, then the above Hamiltonian (61) takes the following form,

$$H^m_{MBL} = xp_z + \frac{\sqrt{z}}{36\beta}p_x^2 + \frac{p_\phi^2}{2z^{\frac{3}{2}}} + \frac{3\alpha}{2\sqrt{z}}(x^2 - 4kz) - \frac{18\beta k}{z^{\frac{3}{2}}}(x^2 + 2kz) + Vz^{\frac{3}{2}}, \quad (62)$$

which is identical to $\mathcal{H}^m_{MHF}$ presented in (28).

### 3.2. Scalar–Tensor Theory: Non-Minimal Coupling

Here, again, upon integrating the action (30) by parts, we obtain

$$A_2 = \int \left[ -\frac{3f\dot{z}^2}{2N\sqrt{z}} - \frac{3f'\dot{\phi}\dot{z}\sqrt{z}}{N} + 6fkN\sqrt{z} + \frac{9\beta}{N\sqrt{z}}\left\{ \left( \frac{\ddot{z}}{N} - \frac{\dot{N}\dot{z}}{N^2} \right)^2 + \frac{2k\dot{z}^2}{z} + 4k^2N^2 \right\} + z^{\frac{3}{2}}\left( \frac{\dot{\phi}^2}{2N} - NV \right) \right] dt. \quad (63)$$

Replacing $\dot{z}$ by $\frac{2N}{3}q$ in view of (13), the above action may be cast as

$$A_2 = \int \left[ -\frac{2}{3}fN\frac{q^2}{\sqrt{z}} - 2f'\sqrt{z}q\dot{\phi} + 6fkN\sqrt{z} + \frac{9\beta}{N\sqrt{z}}\left( \frac{4}{9}\dot{q}^2 + \frac{8kN^2q^2}{9z} + 4k^2N^2 \right) + z^{\frac{3}{2}}\left( \frac{\dot{\phi}^2}{2N} - NV \right) \right] dt. \quad (64)$$

The canonical momenta are

$$p_q = \frac{8\beta}{N\sqrt{z}}\dot{q}, \quad p_\phi = -2f'q\sqrt{z} + \frac{z^{\frac{3}{2}}}{N}\dot{\phi}, \quad p_N = 0 = p_z. \quad (65)$$

As before, leaving out the constraint associate with the lapse function, and replacing $\dot{z} = \frac{2N}{3}q$ in view of (13), the Hamiltonian may be cast as

$$
\begin{aligned}
\text{NH}^{nm}{}_{MBL} &= p_q\dot{q} + p_z\dot{z} + p_\phi\dot{\phi} - L \\
&= N\left[\frac{\sqrt{z}}{16\beta}p_q^2 + \frac{2}{3}qp_z + \frac{p_\phi^2}{2z^{\frac{3}{2}}} + \frac{2f'}{z}qp_\phi + \frac{2fq^2}{3\sqrt{z}} + \frac{2f'^2q^2}{\sqrt{z}} - 6kf\sqrt{z} - \frac{8\beta kq^2}{z^{\frac{3}{2}}} - \frac{36\beta k^2}{\sqrt{z}} + Vz^{\frac{3}{2}}\right],
\end{aligned}
\tag{66}
$$

Finally, applying the canonical transformation relationships $q \to \frac{3}{2}x$ and $p_q \to \frac{2}{3}p_x$, we obtain

$$
\text{H}^{nm}{}_{MBL} = xp_z + \frac{\sqrt{z}}{36\beta}p_x^2 + \frac{p_\phi^2}{2z^{\frac{3}{2}}} + \frac{3x}{z}f'p_\phi + \frac{3f}{2\sqrt{z}}(x^2 - 4kz) - \frac{18\beta k}{z^{\frac{3}{2}}}(x^2 + 2kz) + \frac{9x^2f'^2}{2\sqrt{z}} + Vz^{\frac{3}{2}}.
\tag{67}
$$

Clearly, $\text{H}^{nm}{}_{MBL} \cong \mathcal{H}^{nm}{}_{MHF}$, as presented in (32).

### 3.3. Einstein–Gauss–Bonnet–Dilatonic Action

Finally, integrating Equation (35) by parts, we obtain

$$
\begin{aligned}
A_3 = \int \bigg[ &\alpha\left(-\frac{3\dot{z}^2}{2N\sqrt{z}} + 6kN\sqrt{z}\right) + \frac{9\beta}{N\sqrt{z}}\left\{\left(\frac{\ddot{z}}{N} - \frac{\dot{N}\dot{z}}{N^2}\right)^2 + \frac{2k\dot{z}^2}{z} + 4k^2N^2\right\} \\
&- \frac{\gamma'(\phi)\dot{z}\dot{\phi}}{N\sqrt{z}}\left(\frac{\dot{z}^2}{N^2z} + 12k\right) + z^{\frac{3}{2}}\left(\frac{\dot{\phi}^2}{2N} - NV\right)\bigg]dt.
\end{aligned}
\tag{68}
$$

As before, replacing $\dot{z}$ by $\frac{2N}{3}q$ in view of (13), the above action may be cast as,

$$
\begin{aligned}
A_{3q} = \int \bigg[ &-\frac{2}{3}\alpha N\frac{q^2}{\sqrt{z}} + 6\alpha kN\sqrt{z} + \frac{9\beta}{N\sqrt{z}}\left(\frac{4}{9}\dot{q}^2 + \frac{8kN^2q^2}{9z} + 4k^2N^2\right) \\
&- \frac{2q\gamma'(\phi)\dot{\phi}}{3\sqrt{z}}\left(\frac{4q^2}{9z} + 12k\right) + z^{\frac{3}{2}}\left(\frac{\dot{\phi}^2}{2N} - NV\right)\bigg]dt.
\end{aligned}
\tag{69}
$$

The canonical momenta are

$$
p_q = \frac{8\beta}{N\sqrt{z}}\dot{q}, \quad p_\phi = -\frac{2q\gamma'(\phi)}{3\sqrt{z}}\left(\frac{4q^2}{9z} + 12k\right) + \frac{z^{\frac{3}{2}}}{N}\dot{\phi}, \quad p_N = 0 = p_z.
\tag{70}
$$

As before, leaving out the constraint associated with the lapse function, and replacing $\dot{z} = \frac{2N}{3}q$ in view of (13), the Hamiltonian may be cast as

$$
\begin{aligned}
\text{NH}^{GB}{}_{MBL} =\,& p_q\dot{q} + p_z\dot{z} + p_\phi\dot{\phi} - L \\
=\,& N\bigg[\frac{\sqrt{z}}{16\beta}p_q^2 + \frac{2}{3}qp_z + \frac{p_\phi^2}{2z^{\frac{3}{2}}} + \frac{2\alpha q^2}{3\sqrt{z}} - 6k\alpha\sqrt{z} - \frac{8\beta kq^2}{z^{\frac{3}{2}}} - \frac{36\beta k^2}{\sqrt{z}} \\
&+ \frac{2q\gamma'(\phi)p_\phi}{3z^2}\left(\frac{4q^2}{9z} + 12k\right) + \frac{2q^2\gamma'^2(\phi)}{9z^{\frac{5}{2}}}\left(\frac{4q^2}{9z} + 12k\right)^2 + Vz^{\frac{3}{2}}\bigg],
\end{aligned}
\tag{71}
$$

Finally, the set of canonical transformations $q \to \frac{3}{2}x$, and $p_q \to \frac{2}{3}p_x$, allows one to express the Hamiltonian (71) as

$$
\begin{aligned}
\text{H}^{GB}{}_{MBL} =\,& xp_z + \frac{\sqrt{z}}{36\beta}p_x^2 + \frac{p_\phi^2}{2z^{\frac{3}{2}}} + \frac{3\alpha}{2\sqrt{z}}(x^2 - 4kz) - \frac{18\beta k}{z^{\frac{3}{2}}}(x^2 + 2kz) \\
&+ \frac{x\gamma'(\phi)}{z^2}\left(\frac{x^2}{z} + 12k\right)p_\phi + \frac{\gamma'^2(\phi)x^2}{2z^{\frac{5}{2}}}\left(\frac{x^2}{z} + 12k\right)^2 + Vz^{\frac{3}{2}}.
\end{aligned}
\tag{72}
$$

As a result, one finds $\mathrm{H}^{GB}{}_{MBL} \cong \mathcal{H}^{GB}{}_{MHF}$, as presented in (47). In the process, the pathology discussed with regard to the canonical formulation of Einstein–Gauss–Bonnet–dilatonic action in the presence of a higher-order term is also removed.

## 4. Application

It is mentioned in the introduction that canonical formulation is a precursor to canonical quantization. In the absence of a viable quantum theory of gravity, it is suggested to canonically quantize the cosmological equation and study quantum cosmology to extract some ethos of the pre-Planck era. For example, one can explore the Euclidean wormhole solution. Nonetheless, the 'cosmological inflationary scenario' has been developed since 1980 to solve horizon, flatness (fine tuning), structure formation, and monopole problems, single-handedly. Short-lived $(10^{-36}$–$10^{-26})$ s. inflation occurred just after Planck's era and falls within the periphery of 'quantum field theory in curved space-time'. To be more specific, 'inflation is a quantum theory of perturbations on the top of the classical background', so that the energy scale of the background remains much below Planck's scale. Nonetheless, in this context, Hartle [73] suggested that most of the important physics may still be extracted from the classical action, provided the semiclassical wave function is strongly peaked. The reason is that a correlation between the geometrical and matter degrees of freedom is established, and hence the emergence of classical trajectories (i.e., the universe) is expected. Hence, quantization and an appropriate semiclassical approximation must be treated as a forerunner to study inflation.

Canonical quantization and the semiclassical wave function, in connection with the Hamiltonian (67) for non-minimally coupled higher-order theory, are presented in [26], which reduce to the minimally coupled case when the coupling parameter becomes constant [19]. The Hamiltonian operator was found to be hermitian, the standard probabilistic interpretation holds, and the semiclassical wave function was found to be oscillatory with regard to the classical inflationary solution. Inflation has been studied and the parameters have been found to have excellent agreement with the observational constraints [74,75]. Gravitational perturbation has also been studied.

In [29], again, the quantum counterpart of the Hamiltonian (72) in connection with Einstein–Gauss–Bonnet–dilatonic coupled action is presented. The hermiticity of the Hamiltonian operator is proved, a probabilistic interpretation is explored, and the semiclassical wave function is found to be oscillatory with regard to the classical inflationary solution. Finally, we studied inflation and found that the inflationary parameters more or less satisfy observational constraints [74,75]. In a nutshell, the results obtained in [29] are as follows. The quantum equation takes the form,

$$i\hbar \frac{\partial \Psi}{\partial \sigma} = \left[ -\frac{\hbar^2 \phi}{54\beta_0 x} \left( \frac{\partial^2}{\partial x^2} + \frac{n}{x} \frac{\partial}{\partial x} \right) - \frac{\hbar^2}{3x\sigma^{\frac{4}{3}}} \frac{\partial^2}{\partial \phi^2} + \frac{2i\hbar\alpha_0}{\sigma} \left( \frac{1}{\phi^2} \frac{\partial}{\partial \phi} - \frac{1}{\phi^3} \right) - \frac{2i\hbar\gamma_0 x^2}{3\sigma^{\frac{7}{3}}} \left( 2\phi \frac{\partial}{\partial \phi} + 1 \right) + V_e \right] \Psi$$
$$= \hat{H}_e \Psi, \tag{73}$$

where the proper volume, $\sigma = z^{\frac{3}{2}} = a^3$, plays the role of an internal time parameter, and $n$ is the operator ordering index. In the above equation, $\hat{H}_e$ is the effective hermitian Hamiltonian operator, while the effective potential $V_e$ is given by

$$V_e = \frac{3\alpha_0^2 x}{\sigma^{\frac{2}{3}} \phi^4} - \frac{4\alpha_0 \gamma_0 x^3}{\sigma^2 \phi} + \frac{4\gamma_0^2 x^5 \phi^2}{3\sigma^{\frac{10}{3}}} + \frac{\alpha_0 x}{\sigma^{\frac{2}{3}} \phi} + \frac{\lambda^2 \sigma^{\frac{2}{3}} \phi^2}{3x} + \frac{2\sigma^{\frac{2}{3}} \Lambda M_P^2}{x}. \tag{74}$$

The effective Hamiltonian operator is found to be hermitian for $n = -1$, which selects the operator ordering parameter for physical consideration. The standard quantum mechanical probability interpretation also holds. Under a suitable (WKB) semiclassical approximation, the wave function is found to be

$$\Psi = \Psi_0 e^{\frac{i}{\hbar} \left[ -\frac{6\alpha_0 \lambda z^2}{a_0 \phi_0} + 16\gamma_0 a_0^2 \phi_0^2 \lambda^3 \sqrt{z} \right]}, \tag{75}$$

which exhibits oscillatory behavior with regard to the classical inflationary solution $a = a_0 e^{\lambda t}$, where $\alpha_0$, $\phi_0$, and $\gamma_0$ are constants. We also present several sets of inflationary parameters in [29], which show that the spectral index of scalar perturbation and the scalar-to-tensor ratio lie within the range $0.967 \leq n_s \leq 0.979$ and $0.056 \leq r \leq 0.089$, respectively, showing reasonably good agreement with recently released data [74,75]. The e-folding number also remains within the acceptable range $46 < N < 73$, which is sufficient for solving horizon and flatness problems.

**5. Concluding Remarks**

Although initiated two centuries ago, the canonical formulation of the higher-order theory of gravity is particularly non-trivial. In fact, only after probing dilatonic coupled Gauss–Bonnet action was it found that divergent terms play a major role in formulating the correct quantum dynamics of non-linear gravity theory. The scheme is therefore as follows: First, the action is expressed in terms of the basic variable $h_{ij}$; otherwise, if expressed in terms of the scale factor, which is common, some unwanted divergent terms are removed in the process of integration by parts, which are not found under the variation of the action. Next, unless divergent terms are taken care of, the Hamiltonian is found to be different, which is related through canonical transformation, although such a transformation cannot be carried over in the quantum domain due to non-linearity. It is shown that in the case of Einstein–Gauss–Bonnet–dilatonic coupled action in four dimensions, the Hamiltonian is wrong, since it does not reflect the topological invariance of the theory. This proves the importance of divergent terms in higher-order theories. In this respect, the difference between the BL formalism and the MHF is apparent. In fact, the BL formalism produces an identical Hamiltonian as obtained earlier following Ostrogradsky's, Dirac's, or Horowitz's formalisms. However, MHF is a form of the Horowitz formalism, after expressing the action in terms of the three-space curvature and taking care of the total derivative terms under integration by parts. It was shown that following the same route, if Dirac's algorithm is applied, the Hamiltonian becomes identical to the one found following MHF, and one obtains a unique quantum description. Here, we show that the same is true with the BL formalism. In fact, the BL formalism not only bypasses constraint analysis, as in the case of Horowitz's formalism, but also does not require the action to be cast in canonical form with an auxiliary variable, which can be intricate. In a straightforward manner, it establishes diffeomorphic invariance, and therefore is the easiest technique to use for handling higher-order theories.

**Author Contributions:** The problem was suggested by A.K.S., and the computation was carried out by D.S. and extensively checked by A.K.S. All authors have read and agreed to the published version of the manuscript.

**Funding:** This research received no external funding.

**Data Availability Statement:** Not applicable.

**Conflicts of Interest:** The authors declare no conflict of interest.

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
