# Peer review of "Perusing Buchbinder–Lyakhovich Canonical Formalism for Higher-Order Theories of Gravity"

_universe, doi:10.3390/universe9010048_

Round 1
Reviewer 1 Report
Please, see the attached file

Author Response
Reply to the 1st referee’s report:
- We agree that there had been several typos. We are extremely sorry for that. However, the meaning of ‘mistake’ is not clear. We have not found any grammatical error.
- We have handled three different higher-order theories which had been inspected earlier with other formalisms. In future, we shall definitely study more general ones. But, currently, we don’t find any reason to consider those.
- Indeed , but note that , which is a total derivative term, and does not contribute to the field equations. We have now worked out that up to a total derivative term, at the beginning (equations 6 - 8) of section 2 .
- Hamiltonian is the foundation to study physics. Construction of Hamiltonian for suggested modified theories of gravity, itself is an important physical problem, we believe. Still to satisfy referee’s worry in connection with the journal’s strategy, we have added section 4, to demonstrate applicability.

Reviewer 2 Report
This paper studies the canonical formalism for gravity theories with higher order curvature terms. The "standard" technique in analyzing the constraint of gravity actions is based on Dirac's constraint analysis, while this paper explored an alternative formalism, dubbed the Buchbinder-Lyakhovich formalism. The main finding in this paper is that by expressing the action in terms of the space curvature and removing the total derivatives, all pathologies that arise in previous analysis will disappear. The authors also pointed out the role of the boundary terms in constructing the canonical formalism.
I think the results presented in this paper are interesting and useful not only to the study of canonical structure of gravity theories, but also to the broader community of gravity reseaches. Therefore I recommend this paper to be published on Universe, but after the following issues get clarified.
(1) The author emphasized that it is crucial to choose the spatial metric h_{ij} instead of the scale factor as the basic variable. But since h_{ij} is simply a^2\delta_{ij}, what is the essential difference in two choices? The author should make a comment on this point, perhaps in the "Introduction" section.
(2) Some typos or typesetting issues:
(a) The first word in the Abstract: "Ostrogrdski" should be "Ostrogradski".
(b) In the second line of eq. (39), there seems to be a mis-placed subscript "z".
(c) Ref. [13], "Qunatization" should be "Quantization", Ref. [42] "Cononical" should be "Canonical", etc.. (did the authors type the bibliography by hand?)
(d) The organization of the main text should be improved. For example, the last paragraph in the "Introduction" section should be moved to the main text (e.g., Sec. 2) or other places. Also text in Sec. 2 and 3 should be divided into paragraphs, instead of a single paragraph for a whole subsection. It will make the paper more readable.
Author Response
Reply to the 2nd referee’s report:
- Essential difference between the two choices is now explained, at the very beginning of section 3.
- Typos are corrected, along with the typo in equation (39). We usually type bibliography by hand, to avoid errors occurring with symbols and also to maintain the style of the Journal.
- The main text has been re-organized following referee’s suggestion. The very first paragraph of (2.1) must not be split, since it expresses one single compact idea.

Round 2
Reviewer 1 Report
In the revised version the authors addressed the main points raised in the review report. Therefore I believe that now the manuscript is suitable for publication in this journal.